# Estimating the Morbidity of Robot-Assisted Radical Cystectomy Using the Comprehensive Complication Index: Data from the Asian Robot-Assisted Radical Cystectomy Consortium

**DOI:** 10.3390/cancers17071157

**Published:** 2025-03-29

**Authors:** Alvin Lee Yuanming, Fiona Tan Bei Na, Raj Tiwari, Thomas Kong Ngai Chan, Jeremy Yuen-Chun Teoh, Seok-Ho Kang, Manish I. Patel, Satoru Muto, Cheng-Kuang Yang, Shingo Hatakeyama, Kittinut Kijvikai, Haige Chen, Chikara Ohyama, Shigeo Horie, Eddie Shu-Yin Chan, Lui-Shiong Lee

**Affiliations:** 1Department of Urology, Sengkang General Hospital, Singapore 544886, Singapore; alvin.lee.yuanming@singhealth.com.sg (A.L.Y.); fiona.tan@mohh.com.sg (F.T.B.N.); raj.vikesh.p.k.t@singhealth.com.sg (R.T.); thomas.chan.k.n@singhealth.com.sg (T.K.N.C.); 2Department of Urology, Singapore General Hospital, Singapore 169608, Singapore; 3S.H. Ho Urology Centre, Department of Surgery, The Chinese University of Hong Kong, Hong Kong, China; jeremyteoh@surgery.cuhk.edu.hk (J.Y.-C.T.); eddie@surgery.cuhk.edu.hk (E.S.-Y.C.); 4Department of Urology, School of Medicine, Korea University, Seoul 02841, Republic of Korea; mdksh@korea.ac.kr; 5Discipline of Surgery, Sydney Medical School, University of Sydney, Sydney, NSW 2050, Australia; manish.patel@sydney.edu.au; 6Department of Urology, Westmead Hospital, Westmead, NSW 2145, Australia; 7Department of Urology, Graduate School of Medicine, Juntendo University, Tokyo 113-8431, Japan; s-muto@juntendo.ac.jp (S.M.); shorie@juntendo.ac.jp (S.H.); 8Division of Urology, Department of Surgery, Taichung Veterans General Hospital, Taichung 40705, Taiwan; yangck@icloud.com; 9Department of Urology, Graduate School of Medicine, Hirosaki University, Hirosaki 036-8562, Japan; shingoh@hirosaki-u.ac.jp (S.H.); coyama@hirosaki-u.ac.jp (C.O.); 10Division of Urology, Department of Surgery, Faculty of Medicine, Ramathibodi Hospital, Mahidol University, Bangkok 10400, Thailand; kittinut@gmail.com; 11Department of Urology, Renji Hospital, School of Medicine, Shanghai Jiao Tong University, Shanghai 200240, China; kirbyhaige@aliyun.com

**Keywords:** cystectomy, robot-assisted, Clavien–Dindo, comprehensive complication index, morbidity

## Abstract

Robot-assisted radical cystectomy (RARC) can lead to surgical complications. Grading systems, like the Clavien–Dindo classification (CDC), measure only the most serious issue and might not reflect the cumulative burden of complications. We studied whether another tool, the comprehensive complication index (CCI), could provide a more complete picture of complications. Using data from multiple hospitals, we found that the CCI did not significantly improve the predictions of post-operative outcomes, such as longer hospital stays or higher readmission rates, compared to CDC. This is likely because most patients experienced only a few complications. Our findings suggest that while the CCI may have benefits, its usefulness likely depends on the specific surgical setting and patient population.

## 1. Introduction

To assess the morbidity of surgery, the Clavien–Dindo classification (CDC) has been the most widely adopted and validated system [1]. While the CDC is a simple system for classifying surgical complications and is intuitive to apply in the clinic, it is limited in its ability to capture the total burden of operative morbidity arising from sequential procedural-related complications [2,3]. To better estimate the cumulative impact of multiple complications, the comprehensive complication index (CCI) was created to better define post-operative morbidity in patients [4]. The CCI has been studied in other fields of surgery and has been demonstrated to better capture their associated morbidities [5,6]. The studies demonstrated that CCI is useful for capturing the cumulative morbidity of a major surgery like radical cystectomy [7,8,9]. However, its role in defining morbidity in radical cystectomy (RC) remains uncertain. The Asian Robot-Assisted Radical Cystectomy (RARC) Consortium is a collaborative effort involving nine academic centers situated in Asia and Australia [10] and had previously reported the predictive factors for adverse peri-operative outcomes after RARC and the impact of intra-corporeal reconstruction for RARC [11]. The objective of this current study is to investigate the incremental role of the CCI over CDC in defining the morbidity of RARC in the treatment of invasive bladder cancer.

## 2. Materials and Methods

After institutional ethics board approval was obtained, patients undergoing RARC from January 2007 to December 2020 were retrieved from the Asian Robot-Assisted Radical Cystectomy Consortium database [10]. Standardized data-collection forms were used to gather clinicopathological variables for analysis. The variables analyzed included age at surgery, body mass index (BMI), previous tobacco smoking exposure, comorbidities (such as hypertension, diabetes mellitus and ischemic heart disease), American Society of Anesthesiologists physical status scores (ASA), pre-existing renal impairment, presence of hydronephrosis (HN), tumor and nodal stage and utilization of neoadjuvant chemotherapy [12]. Technical details about the RARC surgery have been previously described [10]. The risk factors for adverse peri-operative outcomes such as prolonged LOS, longer TFI, 30-day readmission and higher EBL were previously elucidated using multi-variate binary logistic regression analysis [10].

Post-operative complications occurring within 90 days of surgery were classified using the CDC [13]. To determine the CCI, each singularly recorded peri-operative complication was graded using CDC, assigned a weight of complication based on CDC and summated [4]. The overall morbidity is reflected on the scale from 0 (none) to 100 (death). For the purposes of this study, CCI was classified into 3 groups: (a) CCI = 0, (b) CCI < 75th percentile and (c) CCI ≥ 75th percentile. Area-under-curve (AUC) of receiver operating characteristic curves for both CCI and CDC were compared in predicting adverse peri-operative outcomes of LOS (>14 days), EBL > 350 mL, TFI > 4 days and 30-day readmission. Statistical significance was set at *p* < 0.05. Statistical analyses were made using Statistical Analysis Software (SAS) Version 9.4.

## 3. Results

### 3.1. Patient Characteristics

There were 568 patients identified from the database that underwent RARC from 1 January 2007 to 30 December 2020 in the participating study centers [10]. The median age of our cohort was 67.2 (IQR 60.5–73.6) years. Eighty-five percent were of male gender. The majority (80.2%) of patients had clinical stage T2 or earlier cancer of the bladder, and 8.3% had positive nodes clinically. About a third (30.5%) received neoadjuvant chemotherapy. For urinary reconstruction, 46.0% underwent intracorporeal urinary reconstruction (ICUR) while the remainder underwent extra-corporeal reconstruction (ECUR). The median LOS was 13 days (IQR 9–19 days). The median time to food intake was 4 days (IQR 3–7 days). The median EBL was 350 mL (IQR 200–600 mL). The 30-day readmission rate was 23.4%. There were no deaths occurring within 90 days of RARC. Further patient characteristics and surgical variables are displayed in Table 1.

### 3.2. Post-Operative Complications

Of the 568 patients included, the complication rate was 44.4% (252/568) within 90-days of surgery. The patients with severe complications (CDC ≥ III) comprised 11.6% of the cohort. There was a total of 309 complications reported in these 568 patients (Table 2). Details of these complications were previously published [7]. Briefly, the most common complications included gastrointestinal complications (13.8%) such as ileus (8.7%) and infective complications (19.5%), like urinary tract infections (7.6%) [7]. There were 7.6% of patients with >one perioperative complication. The mean CCI was 10.2 (±13.5) and median CCI was 0 (IQR 0–21). Figure 1 demonstrates the distribution of CCI scores after stratification by the CDC. Our prior publications showed that pre-existing diabetes mellitus, administration of NAC, and OBS creation were significant for predicting the length of stay. History of smoking was associated with prolonged time to solid food intake of more than 4 days. Previous abdominal surgery, preoperative hydronephrosis, and OBS creation significantly predicted the 30-day readmission rate [10].

### 3.3. CCI and Peri-Operative Outcomes

After adjusting for diabetes mellitus, neoadjuvant chemotherapy, orthotopic bladder substitute creation and intra-corporeal urinary reconstruction, the CCI scores ≥ 75th percentile were significantly associated with greater LOS (>14 days) (OR 2.21, 95% CI 1.47–3.31, *p* < 0.001) compared to when CCI = 0 (Table 3). On adjusted analyses, both CCI < 75th percentile and CCI ≥ 75th percentile were associated with greater odds of 30-day readmission (OR 2.67, 95% CI 1.22–5.85, *p* = 0.014 and OR 3.92, 95% CI 2.45–6.29, *p* < 0.001, respectively) (Table 3). CCI was not associated with a higher risk of prolonged TFI (>4 days). Only CCI ≥ 75th percentile was associated with estimated blood loss > 350 mL (Table 3).

### 3.4. Comparison of CDC and CCI in Predicting Peri-Operative Outcomes

There were no significant differences in the AUC between CDC and CCI in predicting LOS (>14 days), TFI > 4 days, 30-day readmission or EBL (Figure 2 and Table 4).

## 4. Discussion

The population in this study comprised patient profiles which would reflect contemporary clinical practice, such as the administration of neoadjuvant chemotherapy in 30% of patients, orthotopic bladder substitute (OBS) creation in 38.9% and the proportion of pT2-pT4 disease of 63%. Data from International Robotic Cystectomy Consortium showed that the utilization of neoadjuvant chemotherapy was 37% and OBS creation rate was 27% [14,15,16]. Importantly, our cohort’s utilization rate of neoadjuvant chemotherapy is comparable to other contemporary series [17,18]. In this cohort, the CCI was significantly associated with prolonged LOS and an increased 30-day readmission rate. It was also equivalent to CDC in terms of the ability to predict LOS and 30-day readmission rates. In another RARC cohort with intra-corporeal urinary diversion (ICUD) and comprising 885 complications occurring in 507 patients, CCI was found to better define post-operative morbidity compared to CDC in 22.6% of individuals. This finding led the authors to suggest that utilizing the CCI would impact sample size calculations in future RC clinical trials when morbidity outcomes were used as study endpoints [19]. A similar finding of the CCI being better for defining post-operative 30-day morbidity was also described in another RARC and IUCD cohort where 24% of the cohort were upgraded to the most severe complication category on the CCI as opposed to the highest CDC grade [20].

In a cohort of predominantly open RC patients, Huang et al. demonstrated that CCI more accurately predicted LOS compared to CDC, and this cohort had 49.6% of individuals with more than one peri-operative complication [21]. In another open RC cohort of 506 patients, the CCI was found to define more accurately those patients with severe complications, especially when all adverse events were included and recorded [22]. However, no patients within this cohort received neoadjuvant chemotherapy, and in a similar proportion of patients with pT2-pT4 disease (64%), only 20% had an OBS created. In addition, a significant proportion of patients (17%) had a cutaneous ureterostomy created for urinary diversion and all patients received pre-operative bowel preparation. These characteristics differ significantly from the patients described in the present study and other contemporaneous RC cohorts [23,24,25], and the impact of such differences on peri-operative complication rate is not clear. Hence, while the incremental value of the CCI was similarly described, the direct applicability of outcomes from this study may be limited.

The limited benefit of CCI over CDC in this study is likely related to a relatively low proportion of patients who had encountered more than one peri-operative complication (7.6%). This statistical outcome is intuitive as the CCI is a cumulated summation of all morbidities. In other urological domains, it has been shown that CCI was not superior to CDC in correlating with LOS for radical prostatectomy and partial nephrectomy cohorts [26]. Another possible reason could be the fact that our cohort had a low number of T3–T4 patients (<20%). As surgical expertise and experience develop, perhaps more complex cases may be performed using robot assistance in the future and be included in our consortium database.

Hence, this study demonstrates that in determining the morbidity of RARC, the routine use of the CCI across cohorts is context dependent and affected by the overall complication rate of the patient population. The disadvantage of the CCI is the need for a specific calculator while the categorization of CDC does not, and therefore the latter has remained popular in routine clinical practice demanding quick turnovers. However, we agree that the CCI provides superior characterization of overall morbidity especially critical in the determination of health services’ research outcomes and the assignment of appropriate resources for bladder cancer treatment. This is especially so in individuals prone to the highest category of complications based on CCI cohorts and provides a focus group where most efforts can be directed towards pre-operative optimization and peri-operative care.

There are several limitations in this study that need to be acknowledged. As our cohort is pooled from several institutions, there may be variations in the reporting standards of post-operative complications. Major complications of grade III and above in CDC are less likely to be misclassified compared to minor complications which may be categorized either as grade I or II. For example, the duration of peri-operative antibiotics may differ between institutions and patients who receive more than 24 h of such therapy may be classified as a grade II CDC complication despite the absence of clinical infection. Both surgeon and hospital case volume have been associated with early post-operative mortality and complication rates [27,28,29,30]. This was not accounted for in our study as our aim was to reflect real-world practice patterns across institutions. Our cohort encompasses multi-national institutions with differing practices in the availability of care in the community by primary healthcare practitioners, and thus may introduce confounders to outcomes such as LOS and 30-day readmission rates. Outcomes like EBL are usually related to the extirpation component of RARC and remain relevant despite differences in surgical technique for the urinary reconstruction. The study period of our cohort also spanned 13 years and clinical practice impacting outcomes may also be evolving. However, this provides real-life data that reflect continuously evolving surgical and peri-operative medicine, thereby making study outcomes relevant to the reader. All our peri-operative outcomes were recorded and classified in a consistent manner so that they do not affect the statistical analysis. As alluded to in our previous publications, all surgeons within this study had reasonable RARC experience beyond the expected learning curves needed.

## 5. Conclusions

In our multi-institutional cohort, the CCI did not provide additional discrimination over CDC, and this is likely to be related to a limited number of complications occurring per individual. Hence, the perceived advantages of CCI over CDC are contextual.

## Figures and Tables

**Figure 1 cancers-17-01157-f001:**
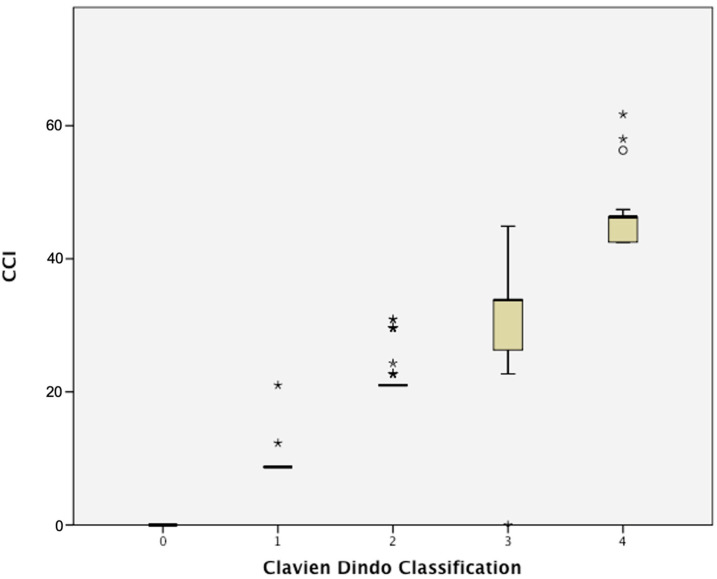
Distribution of CCI scores stratified by the highest Clavien–Dindo complication.

**Figure 2 cancers-17-01157-f002:**
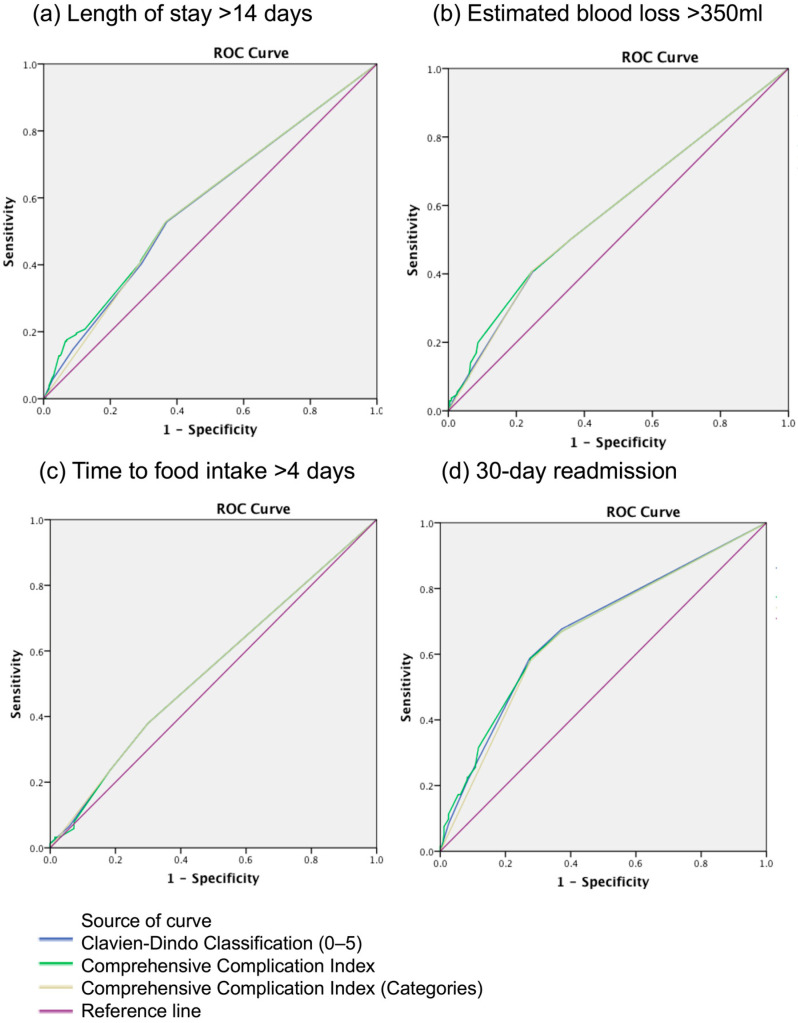
Receiver operating curves for Clavien–Dindo classification and comprehensive complication index for various peri-operative outcomes.

**Table 1 cancers-17-01157-t001:** Baseline demographics of patients undergoing RARC (n = 568).

Pre-Operative Variables	
Median age at surgery, years (IQR)	67.2 (60.5–73.6)
Median BMI, kg/m^2^ (IQR)	24.1 (22.2–26.5)
Male gender, n (%)	484 (85.2)
Previous tobacco smoking exposure, n (%)	282 (50.5)
Comorbidities, n (%)	
Hypertension	243 (42.8)
Previous abdominal surgery	81 (14.3)
Ischemic heart disease	37 (6.5)
Hyperlipidaemia	78 (13.7)
Diabetes mellitus	129 (22.7)
Pre-operative hydronephrosis, n (%)	115 (20.3)
Unilateral	84 (14.8)
Bilateral	31 (5.5)
ASA score, n (%)	
1	102 (18.7)
2	365 (66.8)
3	79 (14.5)
Pre-operative clinical T stage, n (%)	
Tis	21 (3.7)
Ta	25 (4.5)
T1	160 (28.5)
T2	244 (43.5)
T3	85 (15.2)
T4	26 (4.6)
Pre-operative clinical node positive, n (%)	47 (8.3)
Neoadjuvant chemotherapy, n (%)	173 (30.5)
**Intra-operative variables**	
Type of urinary reconstruction	
Intra-corporeal	261 (46.0)
Extra-corporeal	307 (54.0)
Urinary reconstruction	
Ileal conduit	307 (54.0)
Orthotopic bladder substitute	221 (38.9)
Ureterocutaneostomy	33 (5.8)
None *	7 (1.2)
Median estimated blood loss, mL (IQR)	350 (200–600)
Median console time, min (IQR)	345 (266–420)
**Post-operative variables**	
Median time to solid food intake, days (IQR) (n = 290)	4 (3–7)
Median length of hospitalisation, days (IQR)	13 (9–19)
30-day readmission rate, n (%)	133 (23.4)
30-day mortality, n (%)	0
90-day mortality, n (%)	0
Positive surgical margin rate, n (%)	30 (5.3)
Pathological T stage, n (%)	
T0	28 (5.0)
Tis	50 (9.0)
Ta	78 (14.0)
T1	93 (16.7)
T2	113 (20.3)
T3	147 (26.4)
T4	47 (8.5)
Pathological N stage, n (%)	
N0	446 (92.7)
N1	20 (4.2)
N2	8 (1.7)
N3	7 (1.5)
Median lymph node yield, n (IQR)	19 (12–28)
Histology, n (%)	
Urothelial carcinoma	531 (93.5)
Non-urothelial carcinoma	37 (6.5)
Complications, n (%)	252 (44.4)
None	316 (55.6)
Minor (Clavien–Dindo < 3)	186 (32. 8)
Major (Clavien–Dindo ≥ 3)	66 (11.6)
Median CCI, n (IQR)	0 (0–21)
Mean CCI, n (±SD)	10.2 (±13.5)

BMI = body mass index, ASA = American Society of Anesthesiologists Physical Classification, SD = standard deviation; CCI = comprehensive complication index; * Complete urinary tract extirpation or anephric status. Our dataset closely resembles our previously published work on peri-operative outcomes [10].

**Table 2 cancers-17-01157-t002:** Complication rate and prevalence classified by Clavien–Dindo classification within cohort.

Complication	Highest Grade of Complication (Patient Level), n (%)	Number of Complications Occurred, n
Clavien–Dindo 1	55 (9.7)	77
Clavien–Dindo 2	131 (23.1)	165
Clavien–Dindo 3a	20 (3.5)	21
Clavien–Dindo 3b	24 (4.2)	24
Clavien–Dindo 4a	14 (2.5)	14
Clavien–Dindo 4b	8 (1.4)	8
Clavien–Dindo 5	0 (0)	0
Total	568 (44.4)	309

**Table 3 cancers-17-01157-t003:** Predictive factors for adverse peri-operative outcomes.

	Odds Ratio (95% CI)	*p*-Value
**Length of stay >14 days**		
Male gender	1.04 (0.59–1.86)	0.883
Prior tobacco smoking exposure	0.74 (0.50–1.11)	0.144
Diabetes mellitus	1.56 (1.00–2.42)	0.049
Clinical stage T2–T4	0.75 (0.51–1.12)	0.158
Neoadjuvant chemotherapy	2.44 (1.60–3.74)	<0.001
Intracorporeal urinary reconstruction	0.52 (0.35–0.76)	0.001
Orthotopic bladder substitute	2.79 (1.87–4.15)	<0.001
CCI		
0	Reference	
CCI < 75th percentile	1.92 (1.00–3.68)	0.05
CCI ≥ 75th percentile	2.21 (1.47–3.31)	<0.001
**Time to solid food intake >4 days**		
Male gender	1.31 (0.57–3.00)	0.837
Prior tobacco smoking exposure	4.37 (2.39–7.98)	<0.01
Previous abdominal surgery	0.49 (0.18–1.34)	0.165
Clinical node positive	2.48 (0.71–8.75)	0.157
Neoadjuvant chemotherapy	1.77 (0.99–3.18)	0.055
CCI		
0	Reference	Reference
CCI < 75th percentile	1.09 (0.47–2.58)	0.837
CCI ≥ 75th percentile	1.36 (0.68–2.73)	0.383
**Thirty-day readmission**		
Previous abdominal surgery	1.79 (0.99–3.20)	0.051
Pre-operative HN		
None	Reference	Reference
Unilateral	2.20 (1.23–3.94)	0.008
Bilateral	4.04 (1.73–9.46)	0.001
Neoadjuvant chemotherapy	0.28 (0.15–0.50)	<0.001
Orthotopic bladder substitute	1.78 (1.13–2.80)	0.013
CCI		
0	Reference	Reference
CCI < 75th percentile	2.67 (1.22–5.85)	0.014
CCI ≥ 75th percentile	3.92 (2.45–6.29)	<0.001
**Estimated blood loss > 350 mL**		
CCI		
0	Reference	Reference
CCI < 75th percentile	1.05 (0.58–1.90)	0.880
CCI ≥ 75th percentile	2.12 (1.43–3.15)	<0.001

**Table 4 cancers-17-01157-t004:** Comparison of area-under-curve of receiver operating characteristic curves of Clavien–Dindo classification and comprehensive complication index.

Area-under-Curve of ROC Curves	LOS > 14 Days	EBL > 350 mL	Time to Food Intake > 4 Days	30-Day Readmission
CDC	0.581	0.583	0.540	0.674
CCI-categorical	0.579	0.582	0.541	0.664
CCI-continuous	0.588	0.588	0.539	0.675
*p*-value (CDC vs. CCI categorical) *	0.951	0.986	0.990	0.750
*p*-value (CDC vs. CCI continuous) *	0.826	0.876	0.980	0.993

* AUC curves compared using method of Delong; CDC = Clavien–Dindo classification; CCI = comprehensive complication index.

## Data Availability

The datasets presented in this article are not readily available because they are subject to data privacy restrictions from the consortium.

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
