# Peer review of "Estimating the Morbidity of Robot-Assisted Radical Cystectomy Using the Comprehensive Complication Index: Data from the Asian Robot-Assisted Radical Cystectomy Consortium"

_cancers, 2025, doi:10.3390/cancers17071157_

Round 1
Reviewer 1 Report
Comments and Suggestions for Authors
Even if the results do not add anything relevant to the existing literature, the paper is well written and well presented.
The statistical analysis are fine and the presentation of data clear and comprehensive.
Author Response
The authors will like to thank the reviewer for the time taken to review our manuscript.
Reviewer 2 Report
Comments and Suggestions for Authors
The authors present a study regarding the estimation of the morbidity of robotic-assisted radical cystectomy using the Comprehensive Complication Index (CCI). The authors conclude that although in the present study CCI did not provide additional differentiation over the Clavien-Dindo classification (CDC), this is related to the limited number of complications that occurred in the study population and that the perceived advantages of CCI over CDC are contextual.
The structure of the manuscript is correct. The use of English is appropriate, although some sentences require refinement (for example in lines 206-208 and 213-215).
Results
Line 118 : “Details of these complications were previously published”. The authors should elaborate on this topic and provide additional information (most commonly encountered complications , most severe complications etc) about the complications that occurred, so that the presentation of the results to the reader will be more comprehensive.
Line 137-138: “CCI was not associated with a higher risk of prolonged TFI (>4 days) or estimated blood loss > 350ml”. However, in table 3 we can see that CCI ≥ 75 th percentile is significantly associated (p<0.001) with estimated blood loss > 350 ml.
Overall, it is a well-written study providing insights regarding the use of CCI for the estimation of the morbidity of RARC, concluding that the advantage of CCI over CDC is contextual.
Comments on the Quality of English LanguageThe use of English is appropriate, although some sentences require refinement (for example in lines 206-208 and 213-215).
Author Response
Comment 1:
Line 118 : “Details of these complications were previously published”. The authors should elaborate on this topic and provide additional information (most commonly encountered complications , most severe complications etc) about the complications that occurred, so that the presentation of the results to the reader will be more comprehensive.
Response 1: We have included the line, “Briefly, the most common complications included gastrointestinal complications (13.8%) like ileus (8.7%) and infective complications (19.5%) like urinary tract infections (7.6%) [7].”
Comment 2:
Line 137-138: “CCI was not associated with a higher risk of prolonged TFI (>4 days) or estimated blood loss > 350ml”. However, in table 3 we can see that CCI ≥ 75 th percentile is significantly associated (p<0.001) with estimated blood loss > 350 ml.
Response 2: Thank you for pointing this out. The sentence has been corrected accordingly. “CCI was not associated with a higher risk of prolonged TFI (>4 days). Only CCI≥75th percentile was associated with estimated blood loss > 350ml. (Table 3).”
Comment 3:
Overall, it is a well-written study providing insights regarding the use of CCI for the estimation of the morbidity of RARC, concluding that the advantage of CCI over CDC is contextual.
Response 3: The authors would like to thank the reviewer for the time used to review our manuscript.
Comment 4:
The structure of the manuscript is correct. The use of English is appropriate, although some sentences require refinement (for example in lines 206-208 and 213-215).
Line 206-208: There may also be a difference in the surgical technique of urinary reconstruction of the RARC procedure although EBL and is largely related to the extirpation component of RARC.
Corrected to:
Outcomes like EBL is usually related to the extirpation component of RARC and remain relevant despite differences in surgical technique of the urinary reconstruction.
Line 213 – 215: As previously alluded to in our previous publications, all surgeons had in this study had reasonable RARC experience beyond the expected learning curves needed.
Corrected to:
As alluded to in our previous publications, all surgeons within this study had reasonable RARC experience beyond the expected learning curves needed.
Reviewer 3 Report
Comments and Suggestions for Authors
COMMENTS TO CANCERS2025.17
The article is interesting because robot-assisted surgery is currently being imposed.
Compare which scale is best for measuring complications after robotic-assisted cystectomy.
It is very appropriate to decide how to measure complications in order to compare surgical techniques.
It is an article with a clear and pertinent message.
It deserves to be published in its current form.
Author Response
The authors would like the thank the reviewer for the time taken to review our manscript.
Reviewer 4 Report
Comments and Suggestions for Authors
Morbidity of robot-assisted radical cystectomy is an important issue and the authors have addressed this by comparing 2 complications scores. The presentation of their results is clear and their study well defined and established.
However, the cci score needs to be presented in more detail, since it is not as widely used as is the Clavien- Dindo score.
Furthermore, some of the criteria used can not represent the expected worldwide standards. For example using 14 days as a cut-off of the length of stay. In high volume cystectomy centers the length of stay for robotic cystectomy is around 7-10 days.
More over, the data for Clavien Dindo complications are not expected. For example it is unusual to see more cdc 2 than cdc 1 complications and no deaths at all in a total of more than 500 cystectomies within 90 days post op.
Potentially it could be suggested that the comparison should be done with databases from worldwide high volume cystectomy centers.
With the current data, the results presented in the study do not affect nor assist robotic surgeons in their clinical practice.
Author Response
Comment 1:
Morbidity of robot-assisted radical cystectomy is an important issue and the authors have addressed this by comparing 2 complications scores. The presentation of their results is clear and their study well defined and established.
However, the cci score needs to be presented in more detail, since it is not as widely used as is the Clavien- Dindo score.
Response 1: We have added some details as to how CCI is calculated and interpretated.
“To determine the CCI, each singularly recorded peri-operative complication was graded using CDC, assigned a weight of complication based on CDC and summated [4]. The overall morbidity is reflected on the scale from 0 (none) to 100 (death).”
Comment 2:
Furthermore, some of the criteria used cannot represent the expected worldwide standards. For example using 14 days as a cut-off of the length of stay. In high volume cystectomy centers the length of stay for robotic cystectomy is around 7-10 days.
Response 2: The authors acknowledge that worldwide standards do differ in terms of post-operative outcomes such as length of stay. We have acknowledged this in our discussion.
“Our cohort encompasses multi-national institutions with differing practices in availability of care in the community by primary healthcare practitioners, and thus may introduce confounders to outcomes such as LOS and 30-days readmission rates.”
Comment 3:
Moreover, the data for Clavien Dindo complications are not expected. For example it is unusual to see more cdc 2 than cdc 1 complications and no deaths at all in a total of more than 500 cystectomies within 90 days post op.
Response 3: Prior studies evaluating RARC and post-operative complications also show a similar outcomes for CD distribution. The study by Abisinni [1] showed that there were more CD II (29%) complications than CD I (7%). Another study by Mendrek [2] and colleagues also show the similar rates (CD 1 = 16% and CD II = 27). The authors recognise this to be slightly unexpected but our data is congruent with current real-world outcomes and reporting.
Comment 4:
Potentially it could be suggested that the comparison should be done with databases from worldwide high volume cystectomy centers.
With the current data, the results presented in the study do not affect nor assist robotic surgeons in their clinical practice.
Response 4: We thank the reviewer for his insightful comment. We have discussed the complication rates and their risk factors in our prior published work [3]. The main aim of our study was to evaluate the CCI as a marker of cumulative morbidity in patients undergoing RARC. We have discussed the value of CCI across other RARC cohorts in our discussion section. While we do acknowledge that reporting of complications may differ across institutions, the authors have tried our best to standardise complication reporting and grading in our study.
[1] Albisinni S, Diamand R, Mjaess G, et al. Defining the Morbidity of Robot-Assisted Radical Cystectomy with Intracorporeal Urinary Diversion: Adoption of the Comprehensive Complication Index. J Endourol. 2022;36(6):785-792. doi:10.1089/end.2021.0843
[2] Mendrek M, Witt JH, Sarychev S, Liakos N, Addali M, Wagner C, Karagiotis T, Schuette A, Soave A, Fisch M, Reinisch J, Herrmann T, Vetterlein MW, Leyh-Bannurah SR. Reporting and grading of complications for intracorporeal robot-assisted radical cystectomy: an in-depth short-term morbidity assessment using the novel Comprehensive Complication Index®. World J Urol. 2022 Jul;40(7):1679-1688. doi: 10.1007/s00345-022-04051-x. Epub 2022 Jun 7. PMID: 35670880.
[3] Lee AY, Allen JC Jr, Teoh JY, Kang SH, Patel MI, Muto S, Yang CK, Hatakeyama S, Zhang R, Kijvikai K, Chen H, Ohyama C, Horie S, Chan ES, Lee LS. Predicting perioperative outcomes of robot-assisted radical cystectomy: Data from the Asian Ro-bot-Assisted Radical Cystectomy Consortium. Int J Urol. 2022 Sep;29(9):1002-1009. doi: 10.1111/iju.14937. Epub 2022 May 25. PMID: 35613922.
Round 2
Reviewer 4 Report
Comments and Suggestions for Authors
The authors have adressed the main points from the initial review.